# The influence of personality on psychological safety, the presence of stress and chosen professional roles in the healthcare environment

Kate Grailey[1]*, Adam Lound[2], Eleanor Murray[3], Stephen J. Brett[1]

1 Department of Surgery and Cancer, Imperial College London, London, United Kingdom, 2 Patient Experience Research Centre, School of Public Health, Imperial College London, London, United Kingdom, 3 Said Business School, University of Oxford, Oxford, United Kingdom

* k.grailey18@imperial.ac.uk

**Data Availability Statement:** All relevant data are within the paper and its supporting information files. Full interview transcripts containing all

## Abstract

Healthcare teams are expected to deliver high quality and safe clinical care, a goal facilitated by an environment of psychological safety. We hypothesised that an individual's personality would influence psychological safety, perceived stressors in the clinical environment and confer a suitability for different professional roles. Staff members were recruited from the Emergency or Critical Care Departments of one National Health Service Trust. Qualitative interviews explored participants' experiences of personality, incorporating quantitative surveys to evaluate psychological safety and perceived stressors. The 16 Primary Factor Assessment provided a quantitative measure of personality. Participants demonstrated mid-range scores for most personality traits, highlighting an ability to adapt to changing environments and requirements. There was a signal that different personality traits predominated between the two professional groups, and that certain traits were significantly associated with higher psychological safety and certain perceived stressors. Personality was described as having a strong influence on teamwork, the working environment and leadership ability. Our analysis highlights that personality can influence team dynamics and the suitability of individuals for certain clinical roles. Understanding the heterogeneity of personalities of team members and their likely responses to challenge may help leaders to support staff in times of challenge and improve team cohesiveness.

## Introduction

### Background

Personality can be defined as "*the combination of characteristics or qualities that form an individual's distinctive character*" [1]. This refers to the pattern of thoughts, feelings, social adjustments and behaviours which are consistently exhibited over time by an individual. It is these behaviours, thoughts and feelings that contribute to making an individual unique. The

qualitative data are available from the corresponding author on reasonable request.

**Funding:** KG received an unrestricted educational grant from BUPA Cromwell Hospital

**Competing interests:** The authors have declared that no competing interests exist

healthcare environment is complex with multiple competing interests, time pressures and a strenuous workload. Large teams must work together to achieve exemplary patient care, and it is likely that personality will play a significant role in team dynamics–particularly when individuals are under pressure.

Trait theory is an accepted explanation for how personality can manifest in individuals. This theory centres upon the identification, measurement and description of specific traits, subsequently highlighting differences between individuals [2]. A broad based descriptive theory of personality was developed in 1970 by Raymond Cattell, leading to the development of the 16 Primary Factor Questionnaire (16PF) [3]. This questionnaire is a well validated assessment providing scores for 16 distinct personality traits (each originally identified using a factor analysis technique [4]). The stability of personality traits over the course of a lifetime is widely debated. Some studies argue consistency in traits over time, with no change resulting from an individual's interaction their environment [5, 6]. Others postulate that personality traits may demonstrate some dynamic change over the course of a lifetime [7, 8]. Whilst it is not known the extent to which an individual's environment may shape their personality, it is likely that this relationship is predominantly uni-directional, with some personalities being more suited to certain environments, as even when described as dynamic, the traits themselves appear to become more developed or refined, rather than a complete change in personality leading to the manifestation of different behaviours.

The 16PF assessment has been used in healthcare settings (notably with nursing staff and medical student populations) to investigate the impact of personality on measurable outcomes. A study of 159 nurses in 2012 [9] found that high scores for 3 of the 16 primary factors assessed were present in those deemed clinically excellent, and the authors subsequently argue in favour of recruitment to such roles based upon personality. A survey exploring opinions on personality in anaesthetists in Scotland and New Zealand [10] found the majority did not believe personality traits influenced their response to challenging situations, but if presented with the personality traits of potential new recruits that this would influence their hiring decisions.

Several studies of medical students have found an association between the presence of dominant personality traits and performance (both academic ability and non-cognitive skills such as empathy) [11–14]. A study of anaesthetists, using a modified version of the 16PF assessment found significant differences in the personality types of male and female anaesthetists.

Healthcare professionals are subject to a variety of stressors (including high patient acuity and inadequate resources), which can vary depending upon the nature of clinical work [15–17]. It is plausible that different individuals will perceive and respond differently to stressful scenarios within the workplace and that this might be influenced by their personality.

Understanding the stress experienced by healthcare professionals is important when designing interventions to minimise its impact. There is an established link between the presence of workplace stress and increased intention to leave, making workforce sustainability challenging [18, 19].

For teams to function optimally within a stressful environment, the presence of psychological safety is increasingly regarded as paramount [20]. Psychological safety is defined as "a shared belief held by members of the team that the team is safe for interpersonal risk taking" [21]. This can manifest in many ways, including speaking up to prevent error or proposing new ideas without fear of negative personal repercussions. In the healthcare environment this can contribute to quality of care (through the development of new care pathways and policies) and patient safety (through minimising error and near misses) [22].

Psychological safety can be influenced by personality, and positively impacted when the personalities of team members align–(for example, if individuals within the team are all inclined to be proactive, this will promote speaking up behaviours) [23]. An individual's

personality traits can affect their perception of psychological safety—a study of 475 teachers investigated correlations between self-reported psychological safety and traits on the 16PF, finding a significant correlation between high levels of psychological safety and two particular traits (*agreeableness* and *emotional stability*) [24]. Edmondson postulates that personality influences psychological safety at the individual level–but that its impact may be more important early in a team's life–once team members know each other better the impact of personality may lessen [20].

## Rationale for study

Despite differences in role-related demands and the presence of unique stressors within particular clinical departments, the broad person specifications [25] and job requirements are often similar for these roles. This often means individuals self-select for the clinical roles they ultimately find themselves in. Whilst it is challenging to use factors such as personality assessments in recruitment when staff resources are extremely limited, having a greater understanding of the personality types of the individuals working within each clinical team may facilitate team leaders and managers in supporting staff in a bespoke manner by understanding their individual potential response to stress. This in turn may improve job satisfaction, career longevity and ultimately improve patient safety.

The design of this study was informed by previous work by this research group on cognitive processing and personality within the healthcare environment, exploring the response of critical care staff to organisational challenge. A theme highlighting the presence of perfectionist and pragmatic tendencies emerged during the qualitative analysis of an interview study exploring participant's response to a fictional difficult staffing scenario [26]. These different behavioural tendencies and their potential relationship to personality traits were explored further. A secondary analysis demonstrated evidence that certain personality factors were more predominant in this group of critical care staff and differed from the profiles of general population norms. This study also signalled that individuals within this participant group responded to the same clinical stressors in different ways [27]. These signals justified further exploration, both within a larger sample of healthcare workers and expanding to include other clinical specialties.

We aimed to build upon existing knowledge regarding personality in healthcare workers by directly linking personality to the presence of phenomena such as psychological safety, and by evaluating this across two professional groups. We hypothesised that the emergency department–an unpredictable and dynamic setting would attract different individuals to those working in the more controlled environment of critical care. Personality was also explored using two methodologies–a more traditional quantitative assessment, combined with individual perceptions of the influence of personality. As such, this study was designed to explore four hypotheses:

1. There are identifiable differences in the personalities of those working in critical care in comparison to those working in the emergency department.

2. Individuals working in the healthcare environment have distinct personality profiles when compared to general population samples [28].

3. There is a relationship between an individual's predominant personality traits and the clinical situations they perceive as stressful.

4. Higher levels of psychological safety as perceived by participants are associated with predominant personality traits.

## Methods

Mixed methodology was utilised to address the research aims, as this would provide a quantitative assessment of the phenomena studied (such as the perception of psychological safety), whilst also obtaining a qualitative description of participant's individual experiences. Quantitative data regarding an individual's personality traits were obtained using the validated 16PF assessment. The 16PF assessment was selected for several reasons. Firstly, it has frequently been used in similar populations of healthcare workers. Secondly, the breadth of primary factors assessed within 16PF was felt to be useful given the complexity of the healthcare environment. Thirdly, it allowed us to build upon previous work on personality within this research group and enable comparison of traits across similar participant populations. The influence of personality on teamwork and stress as *perceived* by participants was explored using semi-structured qualitative interviews. In addition, two quantitative measures of psychological safety and clinical stressors were included within the qualitative interviews.

The study was reviewed and approved by the Imperial College Research Governance and Integrity Team (Reference number: 19HH5394), Imperial College NHS Trust and the Health Research Authority (Reference number 19/HRA/4541). Adaptations were made to the study methodology and data collection due to disruptions caused by the COVID-19 pandemic. Non-substantial amendments were submitted and approved as required. Formal training in the administration and interpretation of the 16PF personality assessment and candidate feedback was undertaken by KG prior to the onset of study recruitment.

Staff members working at all levels of seniority and all members of the multi-disciplinary team within the emergency and critical care departments of one large NHS trust in London, U.K. were invited to participate.

Participants were recruited according to a purposive sampling strategy with the aim of obtaining a wide range of viewpoints and experiences, thereby reflecting the composition of teams within these clinical environments. This was facilitated by the research teams understanding of the clinical environment [29]. A target sample size of 60 was planned, based upon prior research into the potential influence of personality and previous qualitative studies by this research group [27]. It was anticipated that the duration of recruitment would be determined by a qualitative assessment of thematic saturation, with quantitative data being used to support conclusions made.

All participants were provided with written information prior to their involvement in the study. Once enrolment was confirmed, participants were sent an electronic link to the 16PF assessment, and implied consent was assumed at the point the assessment was accessed. Written informed consent was provided prior to the qualitative interviews.

Recruitment began in September 2019 with an expected duration of 12 months. Study recruitment and data collection were paused for 3 months in March 2020 due to the COVID-19 pandemic, with subsequent changes to the study protocol upon resumption.

Interviews scheduled prior to March 2020 were conducted in person, those performed following the introduction of social distancing restrictions took place via the Microsoft Teams platform. Semi-structured interviews were designed to explore a participant's experiences of stress, psychological safety, teamwork and how their personality influenced this. Within the interviews participants were asked to rank the pre-identified series of clinical stressors (identified during a literature review prior to study commencement)) in order of those they found most stressful. Edmondson's validated assessment of psychological safety was also

incorporated [30], which required participants to state their agreement with 7 statements. The topic guide for the interviews and quantitative assessments can be viewed in S1 File.

A triangulation approach was employed to analyse these mixed methods data, with interpretations based upon the incorporation of multiple data sources [31]. This triangulation was intended to explore the influence of personality on the clinical role participants had chosen, and how personality might influence perception of stress and psychological safety. In addition, it was anticipated that the qualitative data would provide an understanding of how personality might affect teamwork and working relationships.

Qualitative interview data were analysed continually during data collection as an iterative process. This contributed to the development of themes and an assessment of thematic saturation, defined as the point at which no new codes were added to the thematic framework [32, 33]. It was upon reaching this point that recruitment to the study ceased.

Audio interview files were transcribed with all personal identifying information removed and analysed using a thematic analysis technique. Qualitative data were organised and analysed within NVivo Mac R1 software (QSR International Pty Ltd. (March 2020)). The process of thematic analysis was performed primarily by one researcher (KG) in line with published guidance [34] using an inductive approach. The stages of the thematic analysis included familiarisation with the data set, ongoing review and generation of initial codes, searching across the data set for themes and the construction of a thematic framework. This was a recursive process, with ongoing re-review of the original data as the stages progressed.

To confirm analytical interpretations these code and themes were discussed within the wider research team throughout all stages of the analysis. 10% of the interview transcripts were selected at random and cross-coded by a second researcher (AL) to assess for inter-rater reliability. This was done by performing a coding comparison enquiry within NVivo (R1).

Quantitative survey and 16PF data were analysed within Microsoft Excel, with descriptive and inferential statistics investigated using SPSS v27. As data for each 16PF trait is approximately normally distributed within each population group studied, an independent t test was used for the comparison of means between 16PF scores of those in ED and ICU, with Spearman's correlation analysis to explore the relationship and degree of relationship between 'personality and psychological safety' and 'personality and perceived clinical stressors'.

Population data (including groups organised according to both country and profession) is provided in Cattell's 16PF Handbook [28] in the form of mean standard ten (STEN) scores for each trait. Study participant data STEN means were compared with mean STENs for Cattell's original sample of Physicians, Nurses and a British population sample. Whilst we acknowledged that these original sample data were obtained in 1970, given the likely consistency of personality traits over time it was felt that these data would still provide a valid comparison, even though the living and working environments in 1970 and 2020 would have been very different.

## Reflexivity

KG is a PhD Candidate with a background in anaesthesia and critical care, AL is a research physiotherapist, EM is a former NHS manager and is now an academic in organisational studies, SJB is a clinical academic and consultant in critical care. KG, EM and SJB all have previous experience with the conduct and analysis of qualitative and mixed methods studies in the clinical environment. The authors were aware of how their own position may affect the study design, analysis and interpretation of the findings. All authors anticipated there would be an influence of personality on perceived stressors, team dynamics and psychological safety, although had no pre-conceived specific ideas as to how this would manifest. The team

maintained a reflexive position throughout the analysis to minimise the risk that any presumptions would affect the analysis and interpretation of the study findings.

This manuscript is written in accordance with the Standards for Reporting Qualitative Research [35], the checklist for which can be viewed in S2 File.

## Results

Sixty-three participants were recruited, 22 from the two emergency departments and 41 from three critical care units within the Imperial College NHS Healthcare Trust.

All 63 completed the 16PF personality assessment and 58 went on to participate in a semi-structured qualitative interview. The demographics of these 58 participants can be viewed in Table 1. Of these, 49 interviews occurred after the first wave of the COVID-19 pandemic and were conducted virtually. Semi-structured interviews ranged in length from 09:32 to 31:17 with an average duration of 19:46. All planned topics were discussed within each interview, and thematic saturation [32, 36] was achieved within the first 50 interviews.

Six interview transcripts were assessed for inter-rater reliability. Two hundred and forty-six codes relating to the influence of personality on teamwork and the working environment were generated during the thematic analysis and across these the percentage agreement ranged from 99–100%. A kappa coefficient of between 0.4–0.75 (fair to good agreement) was present for 2 codes and value of >0.75 (excellent agreement) in 214.

### Personality profiles

Raw scores for each factor within the 16PF questionnaire are converted to Standard Ten scores ('STEN') and provided for each candidate. This STEN score represents a point on a continuum where an individual's personality sits in comparison to the scores of the wider population. A participant's position on the scale is referred to as low if they score <3, high if they score >8 and midrange for scores between 4–7. Those with either high or low scores will demonstrate a tendency to behave in an extreme manner for each trait.

When analysed as a whole group; most participants possessed midrange scores for each factor–demonstrating an ability to move between the two styles of behaviour at each end of the scale depending upon the requirements of their environment. Of note, there was a marked negative skew towards low scores for *vigilance*, corroborating previous data by this research group on 16PF scores in critical care staff [27]. A tendency for lower scores was also noted for *tension*, *abstractedness*, *privateness* and *self-reliance*. Similarly, candidates tended to score at

**Table 1. Demographics of participants according to department, profession and seniority.**

| Clinical Department | Number of Participants | Professional Group | Number of Participants | Level of Seniority | Number of Participants |
|---|---|---|---|---|---|
| Emergency Medicine | 19 | Nurse | 4 | Junior | 3 |
| | | | | Senior (Band 7 and above) | 1 |
| | | Doctor | 15 | Junior | 10 |
| | | | | Senior (Consultant Level) | 5 |
| Critical Care | 39 | Nurse | 24 | Junior | 18 |
| | | | | Senior (Band 7 and above) | 6 |
| | | Doctor | 9 | Junior | 3 |
| | | | | Senior (Consultant Level) | 6 |
| | | Physiotherapist | 6 | Junior | 4 |
| | | | | Senior (Band 7 and above) | 2 |

the higher end of the scale for *openness to change*. The distribution of 16PF scores for each trait are illustrated in Fig 1.

When analysed according to the clinical department in which an individual participant worked some differences in the personality profiles of each group were demonstrated. A comparison of means for each trait of those working between the emergency department and critical care was performed using an independent t test. A statistically significant difference in means was seen for both *dominance* and *emotional stability*, with participants from the emergency department having higher scores for both (p = 0.017 and 0.026 respectively, and significantly lower scores for *apprehension* (p = 0.024)). Raw data and statistical analyses for the two groups are available in S3 File.

Data published in Cattell's 1970 16PF handbook [28] include mean STENs for each primary factor (excluding *reasoning*). Raw data for each individual participant in the original cohorts are not available, so a statistical analysis to ascertain whether there was a statistically significant difference in means was not possible, however the *trends* of the different personality profiles can be compared. Comparing our participant sample STEN means with the British population sample, a signal demonstrating higher scores for *warmth*, *emotional stability*, *dominance*, *liveliness* and *openness to change* can be observed. Interestingly, differences in means can be seen between our study participant group and Cattell's sample of Physicians and Nurses, with notably higher scores for *warmth*, *liveliness* and *openness to change*. The STEN profiles for these groups are displayed in Fig 2.

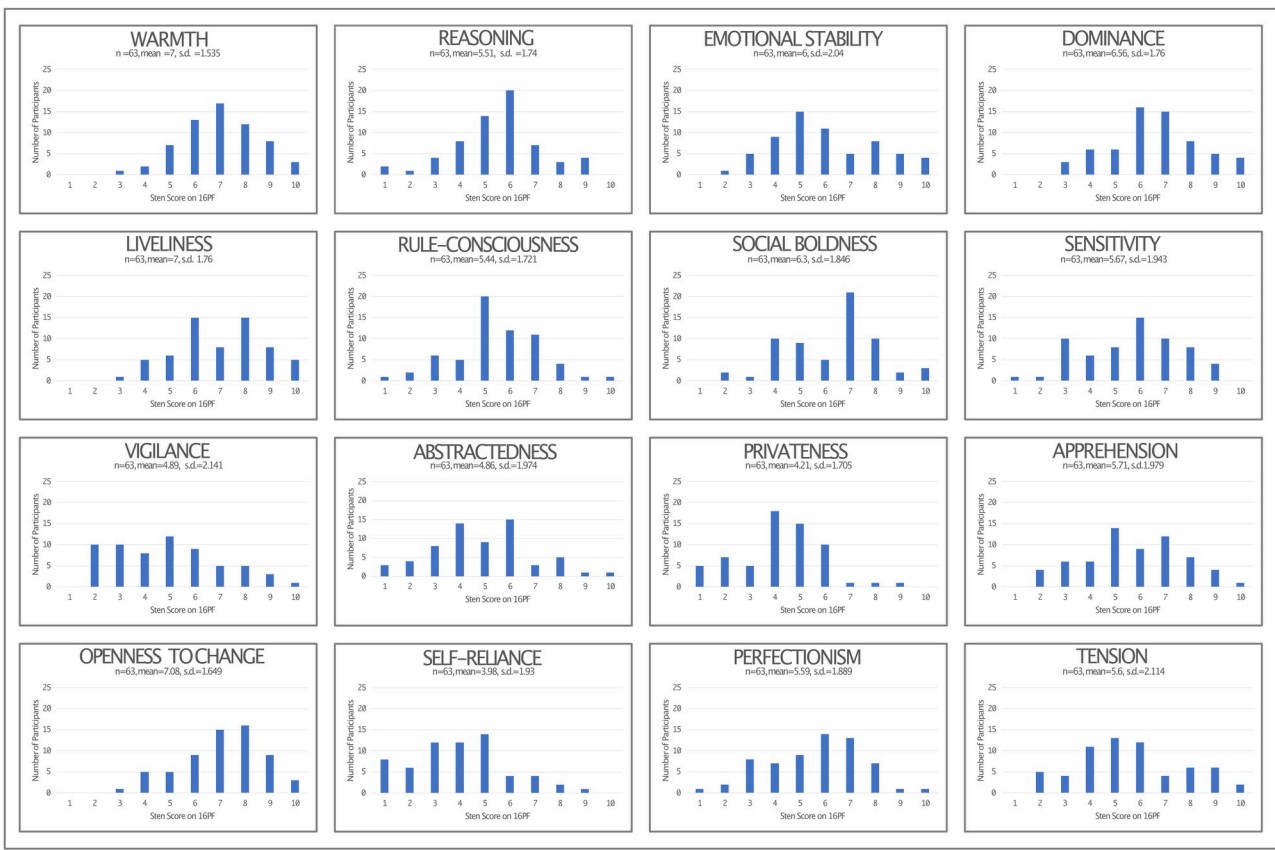

**Fig 1. Distribution of STEN scores on the 16PF assessment for all participants (n = 63).**

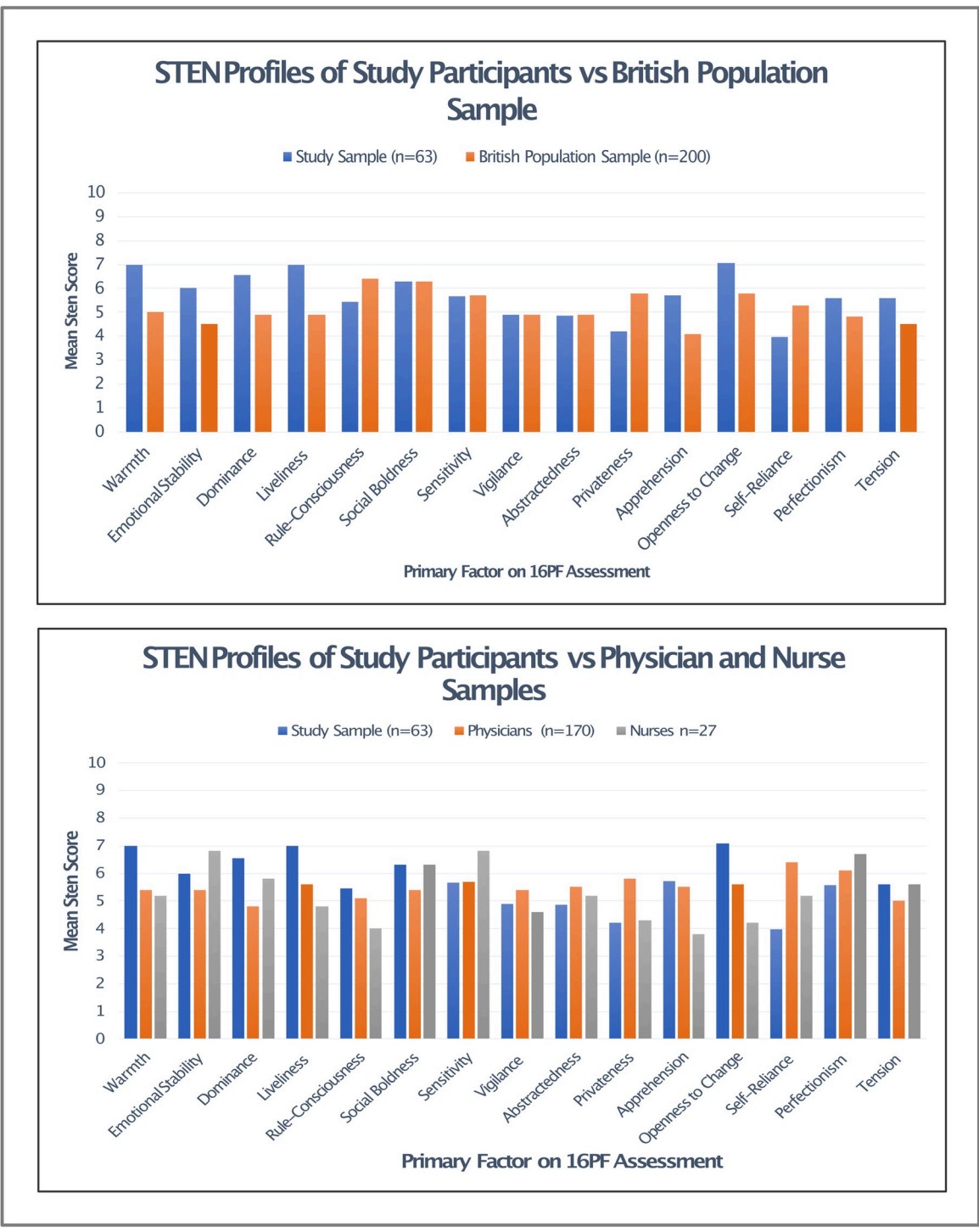

**Fig 2. Comparison between mean STEN profiles for our study population and selected populations within Cattell's 16PF handbook [24].**

**Perceived clinical stressors and relationship to personality traits.** The five most common clinical stressors as identified within the literature review ([37–48]) were '*high workload*', '*patient expectation*', '*conflict with colleagues*', '*beds / resources*', '*risk of making a mistake*'.

High workload was ranked most frequently as being perceived to be the most stressful element of being in a clinical environment, with *'patient expectation'* most frequently being ranked the least stressful aspect. When divided according to clinical department, the patterns of responses were similar, with the exception of *'high workload'* being ranked as most stressful more frequently in critical care staff than by those working in the emergency department. The distribution of responses is illustrated in Fig 3.

Spearman's Rho was used to determine any correlations between STEN scores for an individual's personality traits and which factors they perceived to be the most stressful. For all participants, there were several statistically significant correlations.

Those with high scores for *emotional stability* were more likely to rank *'beds/resources'* as being highly stressful (correlation coefficient -0.293, p = 0.026); those with high scores for *apprehension* were likely to find this factor the least stressful (correlation coefficient 0.272, p = 0.0359).

Those with a higher score for *liveliness* and *social boldness* were more likely to find "*patient expectation*" stressful (correlation coefficient -0.286, p = 0.029, correlation coefficient -0.331, p = 0.011). Individuals with higher scores for *rule-consciousness* were more likely to find "*conflict with colleagues*" stressful (correlation coefficient -0.263, p = 0.046). Raw data and statistical analysis can be viewed in S3 File.

## Psychological safety and relationship to personality traits

Quantitative data relating to individual participant's psychological safety was available for the 58 individuals who participated in the semi-structured interview. These data predominantly demonstrated the presence of psychological safety in both clinical groups. Eighty-eight percent of all participants felt that "members of this team are able to bring up problems and tough issues" and disagreed with the statement "is it difficult to ask other members of this team for help". Lower levels of psychological safety were observed when participants were asked "people

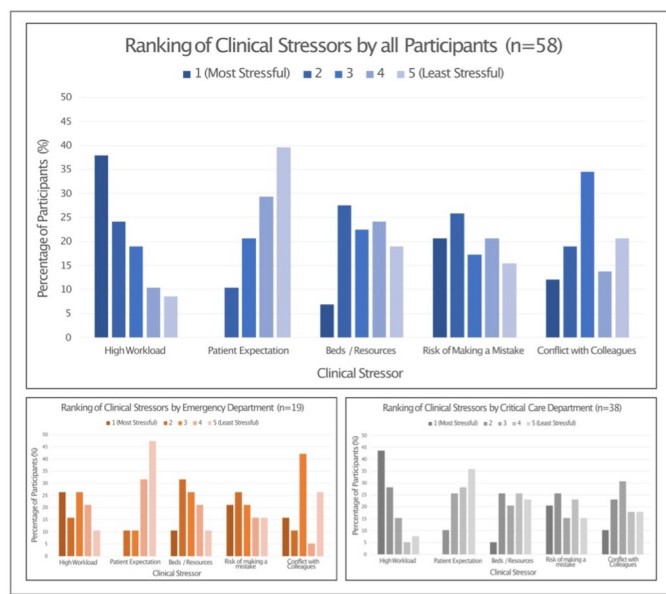

**Fig 3. Variations in ranking of clinical stressors when analysed as all participants, emergency department and critical care.**

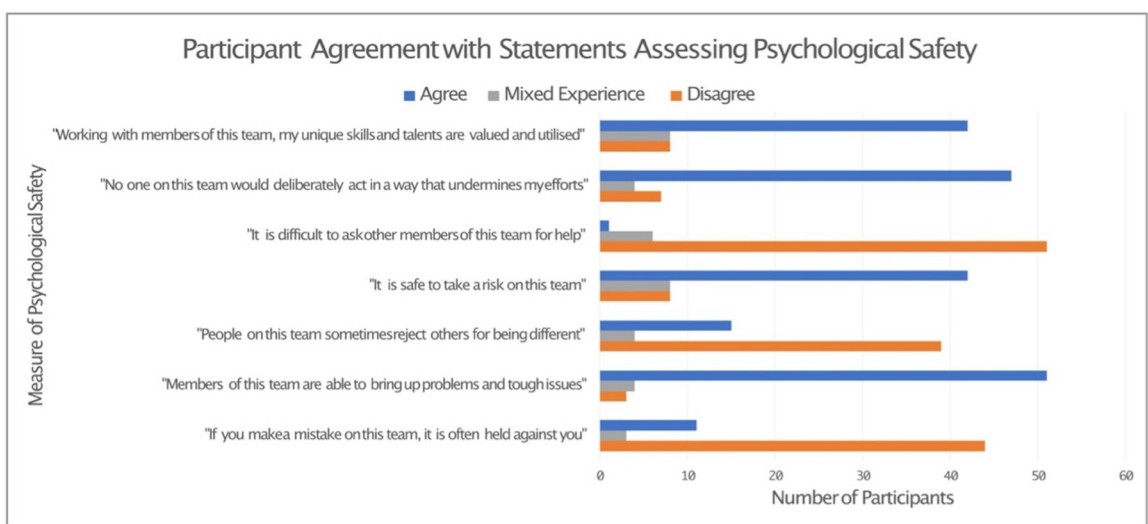

**Fig 4. Participant agreement with measures of psychological safety (Participants contributing both qualitative and quantitative data, n = 58).**

on this team sometimes reject others for being different", with this eliciting only 67% disagreement (Fig 4).

The pattern of responses for this assessment of psychological safety was similar between both clinical departments, with the exception of for "People on this team sometimes reject

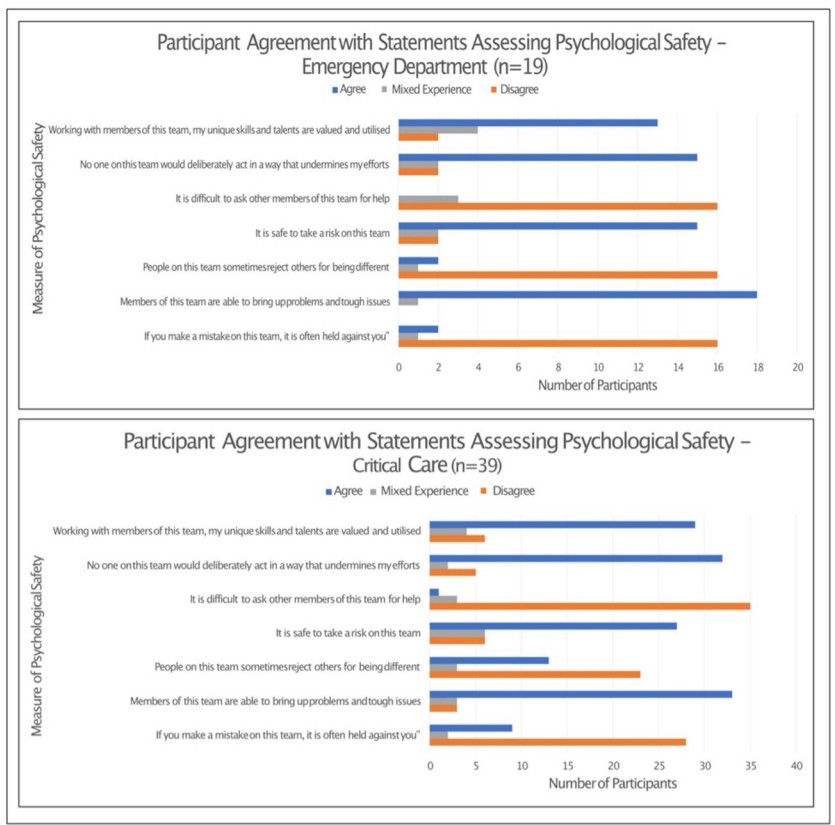

**Fig 5. Participant agreement with measures of psychological safety according to clinical department.**

others for being different". A larger proportion of those working in critical care (33%) agreed with this statement, in comparison to 10% of emergency department staff. The responses according to department can be seen in Fig 5.

A Spearman's Rho statistical analysis was performed to assess for the presence of any correlations between STEN scores for an individual's personality traits and their reported psychological safety. A weak positive correlation was seen between *Emotional Stability* and "no one on this team would deliberately undermine my efforts" (correlation coefficient 0.272, p = 0.039) and, with a weak negative significant correlation between *Abstractedness* and "People on this team sometimes reject others for being different" (correlation coefficient -0.277, p = 0.035).

Statistical data can be viewed in S3 File.

## Qualitative thematic analysis–the influence of personality

Three main themes were constructed during the thematic analysis, with the resulting thematic framework outlined in Fig 6. Further qualitative data are can be viewed in S4 File.

**Theme 1: "Teamwork".**   Personality (as used by participants in a general conversational sense) was felt to have an impact on participants' perception of teamwork in several ways. Sub-themes included a feeling of "fitting in", with participants reporting that their personality had an impact on their ability to work cohesively within the established team–with varying reports of their personality being both beneficial and detrimental in this regard.

> "I think that's a bit of the thing about if you don't fit the mould, then it's harder to fit you into the work environment that everyone else is heading towards" 0035, Consultant, Emergency Department

The impact of personality on team dynamics was frequently discussed by participants, with a sense that the combination of individual personalities within the team had a significant impact upon the way the team operated, communicated and worked together.

---

**INFLUENCE OF PERSONALITY - THEMATIC FRAMEWORK**

**THEME 1: TEAMWORK**
Sub-Themes
- "Fitting In"
- Impact on Team Dynamics
- Conflict
- Psychological Safety

**THEME 2: INDIVIDUAL PERSONALITY TRAITS**
Sub-Themes:
- Impact of one individual on the environment
- Awareness of impact of own personality on team
- Suitability for role
- Individual response to stress

**THEME 3: LEADERSHIP**
Sub-Themes:
- Influence on leadership qualities
- Supporting different personalities within the team

**Fig 6. Thematic framework with major themes and subthemes regarding the influence of personality.**

---

*"Personalities always affect the teamwork, unless you realise that everyone is very different and you accept the fact that everyone is different. Sometimes, people clash, but that's not because anyone wants to clash, it's just because we are different." 0055, Sister, Critical Care*

The contribution of personality to the incidence of conflict was discussed as being significant with many participants, with many feeling that personality was a key causative factor.

*"But I've had a few tiffs with people, but it's been more personality wise." 0086, Staff Nurse, Critical Care*

Participants often agreed that personalities within the team could impact psychological safety. This manifested as being important both in terms of an individual's personality influencing their ability to speak up, but also the personality of those in senior positions who were to be approached.

*"I think often it's a choice of who you would go to and that's probably in terms of personality and how you get on with some people." 0042, Junior Doctor, Emergency Department*

*"I'm not the person who will challenge the senior people. . . But it depends on the consultant as well, it depends on the personality." 0090, Senior Staff Nurse, Critical Care*

**Theme 2: "Individual personality traits".** Many participants perceived an individual worker's personality to be significant for the cohesiveness of the team. This was often highlighted when discussing the impact of one individual upon the environment, and generally related to the perceived stress felt by participants. The data demonstrated that some individuals were able to maintain a calm environment as a function of their personality, whilst others would generate a feeling of stress in those around them.

*"Anybody who's got a stressed personality I feel can influence outcomes because we all need each other within the facility." 0091, Staff Nurse, Critical Care*

It also became apparent that the personality of senior members of staff (notably the consultant or nurse in charge) was perceived to have a significant impact on the atmosphere during the clinical shift.

*"Because you sometimes know if you've got one particular consultant in charge or reg in charge, you just know it's going to be a bit more of a stressful day." 0078, Junior Doctor, Emergency Department*

*"if you've got someone that's calm and like it's going to be fine, you're going to be okay, I'm here if you need anything just let me know, that tends to make them feel instantly relaxed for the shift." 0072, Sister, Critical Care*

A similar but distinct subtheme reflected the participant's awareness of how their own personality could impact the working environment. Participants generally viewed it as having a positive effect—reporting that their inherent calmness was beneficial.

*"And then I definitely noticed I'm normally quite calm. I don't shout at people. I don't tend to get that stressed, and I'm normally quite relaxed." 0072, Sister, Critical Care*

The qualitative data also reflected participants self-perceived suitability for their professional role. Participants discussing how their personality helped them in their role generally did so from a positive perspective, reflecting how they enjoyed the type of work–for example thriving in high pressure situations.

*"I think that's part of the reason I'm in emergency medicine is, it's not about a long lead in and prepping the surgery or anything like that. You've got to respond to the thing in the moment. And so, if there's something about the ability to handle uncertainty." 0052, Consultant, Emergency Medicine*

"I think generally ICU nurses tend to be the type A personality who expects everything to be done to perfection" 0046, Senior Staff Nurse, Critical Care

The data demonstrated how a participant's personality could influence management of stressors. This manifested as observations regarding how others responded to stress–including being risk averse or becoming stressed easily.

*"I'm convinced that some people blossom in under stress and are able to reach out and become much more team players. And then others retreat back into command control mode and I think that varies also with your team." 0069, Consultant, Emergency Department*

*"Yes it is because I see a lot of people worry, and I know that it factors in many people's lives." 0071, Sister, Critical Care*

**Theme 3: "Leadership".** Personality was perceived to influence leadership qualities, with certain observed traits such as calmness or a tendency to collaborate viewed as improving leadership ability.

*"I've definitely seen examples of beneficial traits. Someone that can place themselves very easily in a position of leadership and authority. If you've got an emergency situation, you do need someone to take the lead and take people's first names. Ask them to do specific jobs. Put people into different roles. Lead the team. I think that's all very beneficial." 0079, Junior Doctor, Critical Care*

Participants who were in leadership positions within their clinical environments conveyed that it was useful to know and understand the personalities present within the team in order to improve the working environment and be able to support individuals as needed.

*"because you're such a big group of people you will get personalities that you not necessarily clash with, but you have to be a little bit more patient and diplomatic sometimes." 0080 Staff Nurse, Critical Care*

## Discussion

Our data provide evidence of subtle differences in predominant personality characteristics within the clinical workforce, and valuable insights into the potential impact of this on cohesive team working, the perception of stress and the creation of an environment of psychological safety. We also demonstrate that different personalities can be seen working in different

clinical areas, possibly highlighting that some personality traits are more suited to certain environments than others.

The influence of personality has been explored previously in healthcare workers, but this has been limited, typically investigating the presence of correlations between personality traits and performance [10–14, 49, 50]. These are quantitative studies, and do not assess how personality can influence stress, team dynamics and the working environment. An ethnographic study explored the impact of team dynamics and culture and decision making in surgical teams, but personality did not feature in this analysis [51].

Correlations between personality and psychological safety have previously been explored to a limited degree, but do not have healthcare workers as the predominant participant group. A recent study of miners in Ghana reported that individuals with more "resilient" personalities were naturally more psychologically safe than their counterparts [52]. Another study in miners showed that resilient personality traits (such as emotional stability) would positively predict psychological safety [53]. Other studies have explored how psychological safety can be a moderator in the relationship between proactive personalities and job satisfaction [54], but do not explore the explicit relationship between personality and psychological safety itself.

By combining quantitative measures of personality and psychological safety with a qualitative analysis, the data in this study have allowed a detailed insight into the influence of personality on psychological safety and workplace dynamics in the healthcare environment to be obtained.

## Personality profiles

Participants in this study showed a predominance for mid-range scores across most traits assessed. This is likely to be beneficial for those working in healthcare, allowing them to adapt to the wide-ranging and time-sensitive challenges faced. For *rule-consciousness*, the majority of participants scored 5, meaning a "tendency to accept imposed rules and regulations, but not to do so rigidly, at times turning to what is convenient or practical" [55]. It is easy to see how this could be beneficial within the healthcare setting, with staff preferring to follow protocols and pathways but adapting as needed in the face of resource challenges or emergency situations.

A significant proportion of participants had low scores (<3) for *vigilance*. Rather than conveying some notion of "alertness", this trait relates to the "extent to which people will be wary of and mistrust others", with low scoring individuals "likely to be tolerant and expect fair treatment from others" [55]. Given the need to trust patients, colleagues and those in more senior roles within the organisation, it is not surprising that low scores for this trait predominate in our study sample.

Our data demonstrate a signal that there are differences in predominant personality traits between those working in the emergency department and those in the critical care environment, supporting the first hypothesis. Those in the emergency department had higher scores for *dominance* ("the extent to which the individual wishes to exert an influence on the views, opinions and actions of others") and *emotional stability* ("how calmly a person tends to adapt to the demands life makes upon them") than those working in critical care. The emergency department is dynamic with rapidly changing requirements for those working within it, where higher scores for these traits and therefore subsequent behaviours may be beneficial.

Our sample of study participants showed differences in mean STEN scores for several personality traits when compared to those published within Cattell's 1970 handbook (including higher scores for "warmth", "dominance" and "liveliness", and lower scores for "privateness" and "self-reliance"). Whilst these groups are not directly comparable, both in terms of environment or role, it suggests that those working in acute care environments may have subtly

different predominant personality traits in comparison to the general population This supports the second hypothesis in this study–that individuals working in the healthcare environment have distinct personality profiles when compared to general population samples. There is scope to explore this further, both in terms of whether these predominant traits confer suitability for an individual's professional role, and whether different personality traits are present in individuals working in healthcare specialties not included in this study. It is also worth acknowledging that Cattell's handbook was published in 1970, and the nature of the work performed by professional groups is likely to have changed, alongside the ethnic and cultural profile of both general and healthcare worker populations. Future research would benefit from using contemporaneous comparator population groups.

## Clinical stressors

High workload was ranked 'most stressful' proportionally more in critical care staff, possibly as a reflection of expectation–in comparison to the emergency department where there is expected to be a constant influx of patients.

Five personality traits on the 16PF questionnaire were significantly correlated with the ranking of clinical stressors, and consequently which factors participants found particularly stressful. These data support the third hypothesis, that there would be a relationship between an individual's predominant personality traits and the clinical situations they perceive as stressful. Whilst these data provide only a signal of how personality can influence an individual's perception of their environment, it is useful to acknowledge this. If team leaders can understand the range of personalities within the team, predicting scenarios during which particular individuals could experience high levels of stress may facilitate increased support where needed.

## Psychological safety

Predominantly high levels of psychological safety were demonstrated within the participant group. Confirming the presence of psychological safety facilitated the subsequent exploration of how this phenomenon could be influenced by personality and environment. Psychological safety manifested in different ways between the two departments, demonstrating how factors specific to each department work together to promote (or inhibit) a psychologically safe environment. The correlation of some personality traits with high scores on certain aspects of the psychological safety assessment may suggest that the construction of a psychologically safe environment is influenced by the personalities of individual team members, supporting the fourth hypothesis in this study. This signal also corroborated Edmonson's suggestion that personality can impact a climate of psychological safety [56].

## Qualitative analysis

Our quantitative data provide evidence of differences in personality in our participants and how this might affect perception of stress or psychological safety. By triangulating qualitative data with the quantitative data we are able to explore each participant's lived experience and gain a more detailed understanding of how personality can impact the working environment.

Our thematic framework highlights how the personalities of individuals working together during a shift can combine to create either a calm or a stressful environment in the face of similar clinical stressors. It also identifies that there are key individuals working within the clinical environment, namely those in leadership positions within nursing and medicine whose personality can have a significant impact on the working environment. The benefits of a "calm" leader was a very strong theme within the data, viewed as a positive attribute by participants.

Understanding that personality is influential in the creation of a good working environment and cohesive teamwork is a positive argument for creating teams that have aligned personality traits, or at the very least understanding the personalities present within the team and using this information to improve team cohesiveness.

Our data demonstrate a difference in the strength of certain personality traits present between members of different clinical departments. It is possible to extrapolate from this that selecting for roles within clinical departments may be beneficially assisted using personality assessment tools (both in terms of individual suitability and team cohesiveness). However, given the pressures on the service delivered by the NHS and the continuing staff shortages, departments do not have the luxury of being able to select individuals according to personality type. In addition, performing personality assessments with in-depth candidate feedback requires significant resources including finance, expertise in administering the assessments and time for each participant to undertake the assessment itself. It is also worth considering that heterogeneity of personality within the clinical workforce is likely to be beneficial for clinical outcomes and patient safety, given the potential complexity of clinical medicine and positive effects of diversity; individuals with different personalities may react in different ways to organisational challenges, possibly providing an increased opportunity for problem solving and ingenuity. However, whilst this heterogeneity may be useful, the consequences for the individual if they are particularly vulnerable to stress or mismatched in their place or team of work must also be considered. If these 'at risk' individuals can be identified it may be possible to provide increased support for them in certain challenging scenarios, with the intention of minimising stress, improving job satisfaction and reducing the risk of burnout.

Another application of personality assessments may be for team leaders to use them to "get to know" their team, particularly each individual's potential strengths and weaknesses. As an individual's personality data and predominant traits are unlikely to change significantly over time an individual's personality report can be used as part of ongoing review, particularly in times of challenge to assist in identifying coping strategies and possible reasons for behaviours and responses.

## Future studies

There is opportunity for further research building upon the analyses in this study. It would be beneficial to conduct an in-depth exploration of the relationship between psychological safety and prominent personality traits, and how personalities can influence the creation of a psychologically safe environment. Our qualitative data have also strongly identified themes that illustrate the influence of personality on team dynamics. This can be explored further through ongoing qualitative work, such as ethnographic study–directly observing the interplay of individuals personalities within the clinical environment.

## Strengths and limitations

There are several limitations within this study. Whilst proportional to the size of the departments, the number of participants from the emergency department is smaller than the participant group from critical care, which may impact the strength of conclusions drawn. Of note, the number of nurses recruited from the emergency department was low. Given the timings of study recruitment in this group, this may reflect increased barriers to participation that may have related to the COVID-19 pandemic. This may also impact the strength of the comparisons made between the two departments.

There is a risk of participation bias–as those with an interest in personality theory and its impact on the working environment, or those disproportionately affected by stress within the workplace may have been more inclined to participate in this study.

Within the qualitative data, participants have used the term "personality" in the ordinary conversational sense, rather than according to a strict definition, and this must be considered. As such, this study incorporates two perspectives on personality–a "stricter" quantitative study exploring the presence of certain traits and the relationship of these with defined variables, and a qualitative discussion regarding how individuals perceive "personality" to impact their working relationships. We regard the combination of these two approaches as beneficial–providing a more total description of how personality can influence team dynamics and the working environment.

The statistical analysis performed on quantitative data within the study highlighted some significant correlations between survey and personality traits. However, the large number of variables analysed within each calculation must be considered, as this will increase the likelihood of a statistically significant result occurring by chance. It is also worth noting that whilst our analyses allow for the identification of these correlations, there may have been other key variables (for example, level of clinical experience or duration in a specific role) that may account for the differences in psychological safety or perceived stressors.

## Conclusion

Personality has long been acknowledged as an important part of candidate selection for professional roles [57] with the intention of hiring individuals who will be successful, fit in with the existing team and be suited to the role–the latter crucially minimising an individual's stress and promoting job satisfaction. The usefulness of this approach or the significance of personality is acknowledged within the healthcare literature [58], but rarely used in practice in the healthcare setting. The quantitative data in this study highlight that predominant personality traits can vary between different clinical departments and may influence what an individual finds stressful and how this manifests. This is supported by qualitative data exploring the lived experiences of those working in clinical environments, illustrating how the personalities of individuals within the team can influence perceived stress and team effectiveness. Whilst the use of personality data in healthcare recruitment may be limited by staff shortages, there may be a role for it in coaching individuals deciding their future clinical specialties and working environments, particularly for those earlier in their careers. In addition, understanding personalities of other team members may improve support and team cohesiveness, particularly during times of challenge.

## Supporting information

**S1 File. Topic guide for qualitative interviews–influence of personality.**
(DOCX)

**S2 File. Standards for Reporting Qualitative Research (SRQR) checklist.**
(DOCX)

**S3 File. Statistical analyses.**
(DOCX)

**S4 File. Supporting qualitative data for the influence of personality thematic framework.**
(DOCX)

## Acknowledgments

Infrastructure support for this project was provided by the Imperial Comprehensive Biomedical Research Centre.

## Author Contributions

**Conceptualization:** Kate Grailey, Eleanor Murray, Stephen J. Brett.

**Data curation:** Kate Grailey.

**Formal analysis:** Kate Grailey, Adam Lound.

**Investigation:** Kate Grailey.

**Methodology:** Kate Grailey.

**Project administration:** Kate Grailey.

**Supervision:** Eleanor Murray, Stephen J. Brett.

**Validation:** Kate Grailey.

**Visualization:** Adam Lound.

**Writing – original draft:** Kate Grailey.

**Writing – review & editing:** Kate Grailey, Adam Lound, Eleanor Murray, Stephen J. Brett.

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
