## [Decision Letter · Decision Letter 0]

17 Mar 2023

PONE-D-22-19215The Influence of Personality on Psychological Safety, the Presence of Stress and Chosen Professional Roles in the Healthcare EnvironmentPLOS ONE

Dear Dr. Grailey,

Thank you for submitting your manuscript to PLOS ONE. After careful consideration, we feel that it has merit but does not fully meet PLOS ONE’s publication criteria as it currently stands. Therefore, we invite you to submit a revised version of the manuscript that addresses the points raised during the review process.

We look forward to receiving your revised manuscript.

Kind regards,

Ali B. Mahmoud, Ph.D.

Academic Editor

PLOS ONE

Journal Requirements:

“Infrastructure support for this project was provided by the Imperial Comprehensive Biomedical Research Centre. KG received an unrestricted educational grant from BUPA Cromwell Hospital.”

 “KG received an unrestricted educational grant from BUPA Cromwell Hospital”

Reviewers' comments:

Reviewer's Responses to Questions

**Comments to the Author**

1. Is the manuscript technically sound, and do the data support the conclusions?

Reviewer #1: Yes

Reviewer #2: Yes

2. Has the statistical analysis been performed appropriately and rigorously? 

Reviewer #1: Yes

Reviewer #2: Yes

3. Have the authors made all data underlying the findings in their manuscript fully available?

Reviewer #1: Yes

Reviewer #2: Yes

4. Is the manuscript presented in an intelligible fashion and written in standard English?

Reviewer #1: Yes

Reviewer #2: Yes

5. Review Comments to the Author

Reviewer #1: Thank you for the opportunity to review your interesting article. The research covers an important area - improved interaction between healthcare teams will improve patient care and ensure that patients and their families receive appropriate support. Your paper is clear, and I do not have questions about the academic merit thereof. I have a few text-level suggestions that could improve the flow or language use. For example, Page 4, line 86 healthcare professionals could be made "healthcare settings". Page 6, line 131 "Whilst it is challenging". Page 8 line 172 "a qualitative description". Page 8 line 181, "were included during the qualitative interviews" Page 9 line 200 "It was anticipated that the duration of recruitment would be determined by a ..."Page 11 line 237 "(QSR International Pty Ltd. (March 2020))" (only one bracket is closed presently). Page 20 line 435 "worker's personality"

Other than these very minor suggestions, I think the article is ready.

Reviewer #2: This paper looks at the role of personality in shaping psychological safety in teamwork, perceived stressors, and work setting. The analysis uses a rigorous mixed-methods design, drawing on quantitative data analysis and thematic analysis of in-depth interviews to show that personality characteristics do indeed shape perceived stressors and feelings of psychological safety. The analysis also identifies the importance of personality to team setting, team functioning and work environment. The paper is well-written and easy to read. There are, however, a few places where the methods and conclusions could be clarified. A few revisions for meaning and clarification would be valuable.

First, and probably my main question after reading the manuscript, is the relationship between personality and work environment uni-directional or bi-directional? It seems clear that people working in different areas of the hospital have slightly different personality traits, and that these can shape their stressors and feelings of psychological safety. But surely spending a while working in one environment can actually influence one's personality -- enhancing flexibility for example, or a take-charge attitude, since that is what the job requires. I had a similar question when you compare the findings to those from 1970 -- as a society, wouldn't world events alter collective personality traits, such that we would expect the personality profile of people in 1970 and people in 2020 to be somewhat different? This is not a major change, but I would (a) like to know whether the causal direction here is possibly bi-directional, and (b) what that means for your analysis, including the comparison to the population from the 1970s (and thanks for noting this as a potential limitation at the end -- it might be worth a mentioned earlier). Your second research question asks about the comparison between your population and the general population. Is your only comparison point 1970? If so, is this a valid comparison point?

Second, was there a particular challenge recruiting nurses from the emergency department? The N is quite low, and yet the doctor number was quite high. It would help to know what was the barrier here, and perhaps to think a bit more about what it means. You note in the limitations section that selection bias is likely based on personality, but I am also wondering about selection bias based on work environment (high pressure, difficult personalities, poor-functioning teams). The selection bias based on personality might be consistent across the settings, but selection bias based on work environment could impact your findings more substantially. If this is likely or even possible, it is worth a mention.

Third, there are few places early on the manuscript where you could probably introduce some more specificity without providing too much detail. For example, on page 6 when you discuss the study on teachers, you simply say 'certain traits' influence psychological safety -- what kind of traits? Even specifying a few would be helpful. Similarly on page 25 when discussing the findings you state that people working in acute care may have different personality traits. Can you just specify a few to clarify?

Fourth, is it worth noting as a limitation that your analyses enable you to identify correlations, but not to control for other variables? Is it possible that your correlations may be explained by key variables not controlled for?

Thanks for an interesting paper.

6. PLOS authors have the option to publish the peer review history of their article (what does this mean?). If published, this will include your full peer review and any attached files.

Reviewer #1: **Yes: **Sean Glenton Dicks

Reviewer #2: No

---

## [Author Response · Author response to Decision Letter 0]

28 Apr 2023

We would like to thank the reviewers and editors for their review of our manuscript and for making thoughtful and helpful suggestions. We have edited the manuscript in line with these, without changing the original meaning of the text or interpretation of findings. Please find our responses in bold and in line with the reviewers’ comments below. 

Reviewer #1: Thank you for the opportunity to review your interesting article. The research covers an important area - improved interaction between healthcare teams will improve patient care and ensure that patients and their families receive appropriate support. Your paper is clear, and I do not have questions about the academic merit thereof. 

I have a few text-level suggestions that could improve the flow or language use. For example, Page 4, line 86 healthcare professionals could be made "healthcare settings". Page 6, line 131 "Whilst it is challenging". Page 8 line 172 "a qualitative description". Page 8 line 181, "were included during the qualitative interviews". Page 9 line 200 "It was anticipated that the duration of recruitment would be determined by a ..."Page 11 line 237 "(QSR International Pty Ltd. (March 2020))" (only one bracket is closed presently). Page 20 line 435 "worker's personality"

Thank you very much for your kind review. These changes have been made within the text. The final suggestion relates to a participant quote, so this has not been adjusted as it was perceived that this may alter the intended meaning. 

Other than these very minor suggestions, I think the article is ready.

Reviewer #2: This paper looks at the role of personality in shaping psychological safety in teamwork, perceived stressors, and work setting. The analysis uses a rigorous mixed-methods design, drawing on quantitative data analysis and thematic analysis of in-depth interviews to show that personality characteristics do indeed shape perceived stressors and feelings of psychological safety. The analysis also identifies the importance of personality to team setting, team functioning and work environment. The paper is well-written and easy to read. There are, however, a few places where the methods and conclusions could be clarified. A few revisions for meaning and clarification would be valuable.

Thank you very much for your review of this manuscript. 

First, and probably my main question after reading the manuscript, is the relationship between personality and work environment uni-directional or bi-directional? It seems clear that people working in different areas of the hospital have slightly different personality traits, and that these can shape their stressors and feelings of psychological safety. But surely spending a while working in one environment can actually influence one's personality -- enhancing flexibility for example, or a take-charge attitude, since that is what the job requires. I had a similar question when you compare the findings to those from 1970 -- as a society, wouldn't world events alter collective personality traits, such that we would expect the personality profile of people in 1970 and people in 2020 to be somewhat different? This is not a major change, but I would (a) like to know whether the causal direction here is possibly bi-directional, and (b) what that means for your analysis, including the comparison to the population from the 1970s (and thanks for noting this as a potential limitation at the end -- it might be worth a mentioned earlier). Your second research question asks about the comparison between your population and the general population. Is your only comparison point 1970? If so, is this a valid comparison point?

A) Thank you for raising these extremely interesting points. There are several contrasting ideas within the personality literature, particularly with respect to the question as to whether personality traits are fixed throughout an individual’s life. Some academics have reported that personality traits are consistent and fixed throughout a lifetime (which would mean the relationship between personality and the environment must therefore be uni-directional). Other studies find that traits are more dynamic, which may suggest that it is a more bi-directional relationship. However, given that even the papers postulating dynamic personality traits over time allude to modest changes in personality (for example, they do not suggest significant changes to personality that would result in different behaviours), it is likely that the relationship is still predominantly uni-directional. There is ongoing opportunity for future longitudinal studies to understand the consistency of personality in more depth. This point about the consistency of personality traits over time has been expanded in the introduction (Lines 109-117). 

B) Yes, the only comparison point within this study is 1970, which relates to the availability of large nationwide databases of 16PF data. As far as we are aware, there have been no widespread assessments of the populations’ personality since this (which is a limitation of this study). In line with the theory that personality traits are consistent across a lifetime, we still believe this to be a worthwhile and valid comparison point, although acknowledge that different perspectives on the dynamic nature of personality traits, and the potential for a bi-directional relationship are a limitation within this comparison, as the current world & environments are very different from those experienced in 1970. This perspective and limitation has now been acknowledged earlier in the manuscript (Lines 295-298). 

Second, was there a particular challenge recruiting nurses from the emergency department? The N is quite low, and yet the doctor number was quite high. It would help to know what was the barrier here, and perhaps to think a bit more about what it means. You note in the limitations section that selection bias is likely based on personality, but I am also wondering about selection bias based on work environment (high pressure, difficult personalities, poor-functioning teams). The selection bias based on personality might be consistent across the settings, but selection bias based on work environment could impact your findings more substantially. If this is likely or even possible, it is worth a mention.

Thank you for this important point. There is likely to have been a substantial barrier to participation highlighted by this low number, which may be related to the timing of recruitment in this study. The recruitment period for the emergency department was more disrupted during the COVID-pandemic than was seen for recruitment in the critical care department. There were also significant staffing challenges in the emergency department post COVID which may not have influenced the study results but may have limited willingness to participate in a research study. This low N has now been acknowledged and discussed in the manuscript within the limitations section (Lines 718-722). 

Whilst there was a lower recruitment from the emergency department vs critical care, overall the recruitment was proportional to the size of the environment. During the recruitment process there was no other suggestion that the working environment created any selection bias. 

Third, there are few places early on the manuscript where you could probably introduce some more specificity without providing too much detail. For example, on page 6 when you discuss the study on teachers, you simply say 'certain traits' influence psychological safety -- what kind of traits? Even specifying a few would be helpful. Similarly on page 25 when discussing the findings you state that people working in acute care may have different personality traits. Can you just specify a few to clarify?

Thank you for highlighting this. Specific personality traits have been added to both these sections within the manuscript (lines 157 and 619-620). 

Fourth, is it worth noting as a limitation that your analyses enable you to identify correlations, but not to control for other variables? Is it possible that your correlations may be explained by key variables not controlled for?

Yes, it is very much a possibility that the correlations identified are explained by other variables which have not been controlled for in this study (for example, length of time in a professional role may affect what an individual finds stressful, rather than their personality). This has now been noted as a limitation (lines 739-742). 

Thanks for an interesting paper.

---

## [Decision Letter · Decision Letter 1]

22 May 2023

PONE-D-22-19215R1The Influence of Personality on Psychological Safety, the Presence of Stress and Chosen Professional Roles in the Healthcare EnvironmentPLOS ONE

Dear Dr. Grailey,

Thank you for submitting your manuscript to PLOS ONE. After careful consideration, we feel that it has merit but does not fully meet PLOS ONE’s publication criteria as it currently stands. Therefore, we invite you to submit a revised version of the manuscript that addresses the points raised during the review process.

We look forward to receiving your revised manuscript.

Kind regards,

Ali B. Mahmoud, Ph.D.

Academic Editor

PLOS ONE

Journal Requirements:

Reviewers' comments:

Reviewer's Responses to Questions

**Comments to the Author**

1. If the authors have adequately addressed your comments raised in a previous round of review and you feel that this manuscript is now acceptable for publication, you may indicate that here to bypass the “Comments to the Author” section, enter your conflict of interest statement in the “Confidential to Editor” section, and submit your "Accept" recommendation.

Reviewer #1: (No Response)

Reviewer #2: All comments have been addressed

2. Is the manuscript technically sound, and do the data support the conclusions?

Reviewer #1: Yes

Reviewer #2: Yes

3. Has the statistical analysis been performed appropriately and rigorously? 

Reviewer #1: Yes

Reviewer #2: Yes

4. Have the authors made all data underlying the findings in their manuscript fully available?

Reviewer #1: Yes

Reviewer #2: Yes

5. Is the manuscript presented in an intelligible fashion and written in standard English?

Reviewer #1: Yes

Reviewer #2: Yes

6. Review Comments to the Author

Reviewer #1: Thanks for the opportunity to review this important paper. The improvement of understanding of team dynamic and contributing factors is vital to improving service delivery and staff well-being. My suggestions to improve aspects of your paper are minor.

On page 13, you introduce the 63 participants, and follow immediately with Table 1 which only includes the 58 participants who I assume completed the structured interview. I found this order a little incongruent. The situation can be easily rectified by taking out the line: "The demographics of participants can be viewed in Table 1." from where it is, and after the line: "All 63 completed the 16PF personality assessment and 58 went on to participate in a semi-structured qualitative interview." inserting the line "The demographics of these 58 participants can be viewed in Table 1."

At the end of page 13, there is an apostrophe missing from "A participant's position ...". Similarly, On page 26 "... explore each participant's lived experience ..." needs an apostrophe. Also on page 19: "Many participants perceived and individual worker's personality ... " , worker's needs an apostrophe.

In the direct quote on page 20, "I think generally ITU nurses ..." Should this be ICU nurses?

On page 28: "The number of nurses recruited from the emergency department was low in number". I think the sentence could be ended at "low" rather than having the "in number" at the end.

Finally, the heading to Figure 4 ends with "(All participants, N=58)" Perhaps this should be "(Participants contributing both quantitative and qualitative data, N=58)"

Other than these small comments, I feel the article reads clearly and introduces valuable findings.

Reviewer #2: Thanks for the revisions. They have addressed my questions and, I believe, strengthened the paper. I noticed one typo: page 17. Add an apostrophe to participants perceptions [ie participants' perceptions].

7. PLOS authors have the option to publish the peer review history of their article (what does this mean?). If published, this will include your full peer review and any attached files.

Reviewer #1: **Yes: **Sean Dicks

Reviewer #2: No

---

## [Author Response · Author response to Decision Letter 1]

23 May 2023

We would like to thank the reviewers for their kind and thorough review of this manuscript. The minor changes have been made within the manuscript, with no alteration to the previous analyses or conclusions drawn. Please find our point-by-point response in line below. 

Reviewer #1: Thanks for the opportunity to review this important paper. The improvement of understanding of team dynamic and contributing factors is vital to improving service delivery and staff well-being. My suggestions to improve aspects of your paper are minor.

Thank you. 

On page 13, you introduce the 63 participants, and follow immediately with Table 1 which only includes the 58 participants who I assume completed the structured interview. I found this order a little incongruent. The situation can be easily rectified by taking out the line: "The demographics of participants can be viewed in Table 1." from where it is, and after the line: "All 63 completed the 16PF personality assessment and 58 went on to participate in a semi-structured qualitative interview." inserting the line "The demographics of these 58 participants can be viewed in Table 1."

Thank you for highlighting that this requires clarification. These changes have been made. 

At the end of page 13, there is an apostrophe missing from "A participant's position ...". 

This has been corrected, thank you. 

Similarly, On page 26 "... explore each participant's lived experience ..." needs an apostrophe. 

This has been amended, thank you. 

Also on page 19: "Many participants perceived and individual worker's personality ... " , worker's needs an apostrophe.

Thank you, this has been corrected. 

In the direct quote on page 20, "I think generally ITU nurses ..." Should this be ICU nurses?

The participants interchangeably use ICU and ITU to describe the critical care department. For consistency within this manuscript, this has now been edited to read ICU, with no alteration in meaning of the quote. 

On page 28: "The number of nurses recruited from the emergency department was low in number". I think the sentence could be ended at "low" rather than having the "in number" at the end.

Thank you, this has been amended. 

Finally, the heading to Figure 4 ends with "(All participants, N=58)" Perhaps this should be "(Participants contributing both quantitative and qualitative data, N=58)"

Thank you, this has now been clarified within the manuscript. 

Other than these small comments, I feel the article reads clearly and introduces valuable findings.

Reviewer #2: Thanks for the revisions. They have addressed my questions and, I believe, strengthened the paper. I noticed one typo: page 17. Add an apostrophe to participants perceptions [ie participants' perceptions].

Thank you. This has now been addressed.

---

## [Decision Letter · Decision Letter 2]

24 May 2023

The Influence of Personality on Psychological Safety, the Presence of Stress and Chosen Professional Roles in the Healthcare Environment

PONE-D-22-19215R2

Dear Dr. Grailey,

We’re pleased to inform you that your manuscript has been judged scientifically suitable for publication and will be formally accepted for publication once it meets all outstanding technical requirements.

Kind regards,

Ali B. Mahmoud, Ph.D.

Academic Editor

PLOS ONE

Additional Editor Comments (optional):

Reviewers' comments:

Reviewer's Responses to Questions

**Comments to the Author**

1. If the authors have adequately addressed your comments raised in a previous round of review and you feel that this manuscript is now acceptable for publication, you may indicate that here to bypass the “Comments to the Author” section, enter your conflict of interest statement in the “Confidential to Editor” section, and submit your "Accept" recommendation.

Reviewer #1: All comments have been addressed

2. Is the manuscript technically sound, and do the data support the conclusions?

Reviewer #1: (No Response)

3. Has the statistical analysis been performed appropriately and rigorously? 

Reviewer #1: (No Response)

4. Have the authors made all data underlying the findings in their manuscript fully available?

Reviewer #1: (No Response)

5. Is the manuscript presented in an intelligible fashion and written in standard English?

Reviewer #1: (No Response)

6. Review Comments to the Author

Reviewer #1: (No Response)

7. PLOS authors have the option to publish the peer review history of their article (what does this mean?). If published, this will include your full peer review and any attached files.

Reviewer #1: **Yes: **Dr. Sean G Dicks PhD

---

## [Editor Report · Acceptance letter]

26 May 2023

PONE-D-22-19215R2 

The influence of personality on psychological safety, the presence of stress and chosen professional roles in the healthcare environment 

Dear Dr. Grailey:

I'm pleased to inform you that your manuscript has been deemed suitable for publication in PLOS ONE. Congratulations! Your manuscript is now with our production department. 

Kind regards, 

on behalf of

Dr. Ali B. Mahmoud 

Academic Editor

PLOS ONE